# Two Modes of Th1 Polarization Induced by Dendritic-Cell-Priming Adjuvant in Vaccination

**DOI:** 10.3390/cells12111504

**Published:** 2023-05-29

**Authors:** Tsukasa Seya, Masashi Shingai, Tomomi Kawakita, Misako Matsumoto

**Affiliations:** 1Nebuta Research Institute for Life Sciences, Aomori University, Aomori 030-0943, Japan; matumoto@pop.med.hokudai.ac.jp; 2Department of Vaccine Immunology, Hokkaido University Graduate School of Medicine, Sapporo 060-8638, Japan; 3Division of Vaccine Immunology, Hokkaido University International Institute for Zoonosis Control, Sapporo 001-0020, Japan; shingaim@czc.hokudai.ac.jp (M.S.); tomomi-k@czc.hokudai.ac.jp (T.K.); 4Division of Biologics Development, Hokkaido University International Institute for Zoonosis Control, Sapporo 001-0020, Japan; 5International Collaboration Unit, Hokkaido University International Institute for Zoonosis Control, Sapporo 001-0020, Japan; 6Institute for Vaccine Research and Development (HU-IVReD), Hokkaido University, Sapporo 001-0021, Japan

**Keywords:** Toll-like receptor 3, dendritic cell, Th1 polarization, cross-antigen presentation

## Abstract

Viral infections are usually accompanied by systemic cytokinemia. Vaccines need not necessarily mimic infection by inducing cytokinemia, but must induce antiviral-acquired immunity. Virus-derived nucleic acids are potential immune-enhancers and particularly good candidates as adjuvants in vaccines in mouse models. The most important nucleic-acid-sensing process involves the dendritic cell (DC) Toll-like receptor (TLR), which participates in the pattern recognition of foreign DNA/RNA structures. Human CD141^+^ DCs preferentially express TLR3 in endosomes and recognize double-stranded RNA. Antigen cross-presentation occurs preferentially in this subset of DCs (cDCs) via the TLR3–TICAM-1–IRF3 axis. Another subset, plasmacytoid DCs (pDCs), specifically expresses TLR7/9 in endosomes. They then recruit the MyD88 adaptor, and potently induce type I interferon (IFN-I) and proinflammatory cytokines to eliminate the virus. Notably, this inflammation leads to the secondary activation of antigen-presenting cDCs. Hence, the activation of cDCs via nucleic acids involves two modes: (i) with bystander effect of inflammation and (ii) without inflammation. In either case, the acquired immune response finally occurs with Th1 polarity. The level of inflammation and adverse events depend on the TLR repertoire and the mode of response to their agonists in the relevant DC subsets, and could be predicted by assessing the levels of cytokines/chemokines and T cell proliferation in vaccinated subjects. The main differences in the mode of vaccine sought in infectious diseases and cancer are defined by whether it is prophylactic or therapeutic, whether it can deliver sufficient antigens to cDCs, and how it behaves in the microenvironment of the lesion. Adjuvant can be selected on a case-to-case basis.

## 1. Introduction

Humans sustain their survival via constant exposure to diverse microbial environments. Human populations exhibit heterogeneous reactivity to infections from individual to individual via environmental selection. The host response, rather than the microorganisms themselves, is the main driver of infectious diseases. Vaccines, on the other hand, have been discussed empirically in the context of the “no-twice-infected phenomenon” and “post-Jenner vaccines”. It can be said that the immune response has been established as a mechanism for eliminating microorganisms, but the immune response to vaccines varies from individual to individual in each infectious disease and also to mutant strains. In addition, both excessive responses and adverse reactions induce inappropriate immune aberrance in the host.

Vaccines are being developed primarily as therapeutic and prophylactic vaccines for cancer and infectious diseases. In both cases, cDC priming with safe adjuvants has been proven to be essential. However, non-toxic and versatile adjuvants that can be used in the elderly and patients with underlying diseases have not been established. This review aims to introduce the properties of adjuvants that target cDC particularly in infectious diseases, and refer them to cancer vaccine adjuvants. We focus on nucleic acid adjuvants inducing Th1 polarization required for prophylactic vaccine against intractable infectious diseases. 

Adjuvants have been classified as receptor-defined and undefined. Alum (aluminum salt) has been widely used in licensed vaccines targeting pathogens for humans, but the receptor is still undefined [1]. Saponins regulate the Th1/2 polarization of DCs, but their direct receptors are undefined [1]. Alum vaccines have the advantage of increasing antibody production under Th2 situation, but the causal attribution of adjuvant activity is unknown in alum, implying that no method exists to rescue it from an adverse event in case it occurs. In contrast, nucleic acid adjuvants have defined receptors, such as TLRs, and TLR signaling has been clarified at the molecular level.

Subcomponent vaccines usually consist of an antigen and an adjuvant. An adjuvant is a substance that enhances the immune response against non-self antigens in vertebrates, including humans, and is an essential component of vaccines [1]. Adjuvants (i) modulate the rate, time duration, and quality of the immune response; (ii) maximize the immune response to vaccine antigens; and (iii) induce antigen-specific acquired immunity [2]. Multiple mechanisms exist in the action of adjuvants in a broad sense. Although the molecular mechanism for (i) is not always well understood, it promotes sustained stimulation of the immune system via depot formation, in which the antigen is retained and slowly released [1,3,4]; (ii) and (iii) refer to a narrow-sense adjuvant that can be functionally defined as a ligand–receptor molecular response. It is crucial for adjuvants to promote the recruitment of immune cells and facilitate the maturation of antigen-presenting dendritic cells (cDCs) via the production of cytokines and chemokines [1,3].

Once mature, cDCs acquire the ability to process and cross-present extrinsic antigens on MHC class I molecules and induce a specific immune response against foreign non-self antigens via the proliferation of lymphocytes [1,4]. In this context, adjuvants can modulate the activation mode of DCs to induce a variety of CD4^+^ T cell subsets [4]. Th1 polarization creates a favorable environment for the induction of cellular and humoral immunity. Th2 polarization strongly activates humoral immunity but suppresses cellular immunity. The Th17 shift contributes to defense against infections, and the Th22 shift induces a more specific immune response [4,5]. Th17 and Th22 cells (releasing interleukin (IL)-17 and IL-22, respectively) especially play important roles in defending against fungal and bacterial infections on mucosal surfaces, including the lungs and gut. These are in part attributed to the fact that adjuvants induce different types of cytokines/chemokines [6]. Selecting an appropriate adjuvant for the target antigen (from the pathogen) is indispensable for a successful vaccination. 

Several other functions appear, besides cDC maturation, in antitumor nucleic acid adjuvants. They can act on stroma cells to reprogram the tumor microenvironment (TME) [6,7]. Cytotoxic T lymphocyte (CTL)-attracting chemokines are released in response to RNA adjuvants. However, what happens to adjuvants in the pathogen-associated microenvironment during vaccination remains unsolved. In this review, we introduce the issues related to these adjuvants to prepare preventive vaccines against infectious diseases.

## 2. Alum Adjuvants

Alum adjuvants have been widely used in vaccines against human infectious diseases. Alum adjuvants, when combined with antigens, potentially primarily induce a Th2 response and antibody production, and thus have been used in many routine and voluntary vaccines that successfully protect against pathogens via antibody responses [1,4]. Alum has been an appropriate and safe component of licensed vaccines against pathogens for approximately 80 years. The possible mechanisms for the immune-potentiating effects of alum include antigen depot properties, in which antigens accumulate and are slowly released from immune sites [3,4,8]. Skin irritation and fever caused by alum adjuvants have been reported to be mild, but the mechanism of action of the alum adjuvants remains unknown. Furthermore, alum adjuvants may induce the excessive activation of Th2-polarized immunity, leading to adverse effects such as the promotion of autoimmune diseases [9] and antibody- or vaccine-dependent disease enhancement (ADE/VDE) [10,11]. Moreover, the Th2 shift in acquired immunity promotes tumor cell proliferation and lifestyle diseases [12,13]. On the contrary, alum is ineffective against infections such as *severe acute respiratory syndrome* (COVID-19) and respiratory syncytial virus infections, where T cell immunity is essential for the early elimination of the pathogen [11,14]. Therefore, a new adjuvant is desired that covers both the antigen-specific antibody and cellular responses in Th1 polarization without toxicity.

In this context, we explored the possibility of adapting nucleic acid adjuvants currently used for cancer to be used as vaccines against infectious diseases. These adjuvants act on Toll-like receptors (TLRs) in DCs, and preferentially induce Th1 polarization, referred to as Th1 adjuvants. As the receptors for DNA/RNA in DCs are identified and common, in principle, across humans and mice, the mechanism of DC maturation can be analyzed in mouse models. The function of DC-mediated Th1 skewing can be molecularly defined, laying out the results for mice in humans. The role of Th1 adjuvants in vaccines is, to our understanding, in association with DC maturation and cross-antigen presentation. 

## 3. DCs and TLRs

DCs are short-lived (3−5 days) bone-marrow-derived cells that may be supplied locally from the blood pool at any time, differentiating from myeloid precursors to macrophage–DC progenitors and then to common DC progenitors (CDPs) in bone marrow [15]. CDPs are differentiated into DCs by Flt3L [15,16]. Pre-DCs then migrate from the bone marrow into the blood and differentiate into conventional DCs (cDCs) in secondary lymphoid tissues and other tissues. In contrast, the cells differentiated from CDPs to plasmacytoid DCs (pDCs) in the bone marrow enter the blood and migrate to the tissues [15]. 

The following three types of signals are thought to be simultaneously initiated in activating DCs associated with infection [17,18]. The first comprises inflammatory cytokines such as tumor necrosis factor α released from myeloid cells that infiltrate the infected area [17], the second comprises components derived from dead cells such as neutrophils and macrophages that die as a result of infection (i.e., DAMPs) [19,20], and the third comprises signals from TLRs that recognize components derived from bacteria and viruses (e.g., lipopolysaccharides from Gram-negative bacteria and dsRNA from viruses) [21]. The third signal from TLRs participates in the specific targeting of cDCs, evoking the acquired immune system to eradicate infections [21,22]. The cDCs remain at the site of infection for several hours, take up sufficient antigens and become activated, and then migrate through the lymph vessels to their lymph nodes (LNs), where they activate naive T cells, ending their life span within 1 week [18,23]. The infected lesion is then supplied with new DCs from the bone marrow, and as long as the infection continues, the steps of activation of DCs at the site of infection and migration to the regional lymph nodes are repeated periodically [23]. The life cycle and behavior of DCs completely differ from those of macrophages, which stay in the infectious area. Via migration to local LNs, antigen-presenting DCs specify their function to the cross-antigen presentation of foreign antigens through their maturation [4]. 

CD141^+^ DCs are professional DC subsets for cross-antigen presentation in humans. This DC subsets correspond to mouse DC subsets CD8α^+^ and CD103^+^ [18] and commonly express XCR1 [24]. Thus, XCR1^+^ is a representative marker reflecting the function of cross-antigen presentation in DCs. CD141^+^ DCs are BDCA3-positive, which represents the expression of thrombomodulin protein on the cell-surface [24]. 

CD141^+^ DCs express TLR3 in endosomes, and are upregulated in response to IFN-I [18]. Recent studies have shown that TLR3 in cDCs can be targeted for activation via a synthetic ligand ARNAX or Riboxxim (Figure 1) [25,26]. In contrast, CD1c^+^ DCs express TLR4 on the cell surface, and are activated in response to lipopolysaccharide (LPS) via two arrays of adaptors, MyD88 and TICAM-1 (TRIF) [27,28]. Lipid A is an active center for TLR4. The structure of Lipid A determines the biased activation of either MyD88 or TICAM-1 [29,30]. Previous studies have suggested that monophosphoryl Lipid A (MPLA) predominantly activates the TICAM-1 pathway, allowing it to accomplish a less toxic TLR4 agonist [29,31,32]. Unlike TLR3, TLR4 requires TICAM-2 (TRAM) to link TICAM-1 [30]. TLR4 and TLR3 share the TICAM-1 pathway that leads to IL-12 secretion and Th1 skewing [28,29], and contribute to safe DC priming [31,32,33]. IFN-γ plays a pivotal role in establishing the Th1 shift [32]. MPLA has been used for preparing several preventive vaccines against infectious diseases [34,35]. TLR4 may evoke antiviral immunity [36], but does not recognize nucleic acids. 

Both human and mouse antigen-presenting DCs are XCR1-positive, and express TLR3 in common and respond to ARNAX. However, mouse antigen-presenting DCs, including CD8α+ and CD103+ subsets, express TLR4, while their human counterparts CD141^+^ DCs do not [18]. 

## 4. pDCs and cDCs in Recognition of Nucleic Acid Adjuvants

The human pDC subset possesses the MyD88 pathway for TLR9/7-ligand-stimulated activation [37,38]. pDCs express TLR9/7 in endosomes and recognize the unmethylated CpG DNA and single-stranded RNA of viral species to produce IFN-I [37,38]. The transportation of the TLR complexes into the endosomal compartment requires the adapter AP3 [39]. Rather than viral DNA/RNA, synthetic ligands (such as oligonucleotides (ODNs) and quinoline-derivatives) have been used to elucidate TLR9/7-specific signaling pathways [40] (Figure 1). pDC activation results in the production of robust IFN and inflammatory cytokines, and these cytokines directly eliminate viruses [37,38]. pDCs lack the ability to present antigens [40]. K-type ODNs directly stimulate pDCs and B cells and promote antibody production [41]. TLR9 favors B-cell-mediated antibody production in comparison with TLR3. CpG ODN (but not GpC ODN) stimulation in B cells promotes preB cell differentiation into plasma cells and memory B cells. PreproB (naive B) alters CD5^+^ B cells to produce IL-10 and suppresses IL-12 production by cDCs, hindering Th1 polarity [42]. Furthermore, the robust cytokine storm caused by pDC activation can exacerbate smoldering inflammation and autoimmune disorders.

pDCs release IFN-I/cytokines and secondarily activate antigen-presenting DCs in the local environment. pDCs enhance cDC cross-antigen presentation via released IFN-I/cytokines that are rooted in pDC nucleosensing (Figure 1). Inflammation is an absolute prerequisite for such cDC-driven cross-antigen presentation. Thus, pDCs participate in this “inflammatory mode” of antigen presentation via cDC maturation. As severe inflammation can lead to Th2 polarization, Th1 suppression, and tumor growth [43], ligand formulations have been devised to reduce these side effects in nucleic acid adjuvants. pDC-specific targeting by TLR9/7 agonists is worth developing, but it must be improved in clinical trials without exacerbating diseases. 

In addition, human DCs, including pDCs, respond to R848, a common ligand for TLR7 and TLR8 [44]. Both TLR7 and 8 appear to be activated in response to single-stranded RNAs with different AU/GC content [44]. VTX-2337 is known to specifically trigger TLR8 activation. It is known that TLR8 stimulation promotes an NK-cell-dependent immune response against tumors, which is accompanied by increased IFN-γ production [44,45]. Thus, direct action of NK cells by TLR8 ligands promotes tumor regression [46]. Human-bone-marrow-derived suppressor cells express TLR8. Human DCs, macrophages, and monocytes also express TLR8, and human immature myeloid cells express TLR8 at high levels; the role of DC-specific TLR8 is not clearly demonstrated by the addition of TLR8 agonists to cultured human cells. In addition, it is difficult to obtain results with TLR8 agonists in mice.

In contrast, the human cDC subset CD141^+^ DCs predominantly express TLR3 [47], which recognizes dsRNA and transmits signals via the TICAM-1 adapter to mature cDCs for cross-antigen presentation [48]. This is “a non-inflammatory mode” for direct DC priming. Except for direct cDC infection (such as by the measles virus) [49], viral blunted dsRNAs derived from infected cells barely enter the endosomes of DCs [50,51]. Naked viral dsRNAs thereby induce no DC maturation [50,51]. The belief at present is that exosomes containing dsRNAs may deliver viral dsRNAs to cDC endosomes [52]. However, multiple DNA/RNA sensors localized in the cytoplasm, especially known as the DEAD-box helicase family (or sentinels), may co-work with TLR3 [52,53]. Hence, vaccination and actual infection differ in the aforementioned terms. 

Viral dsRNA entering from the outside is mostly AU-rich and rarely reaches the endosomes where TLR3 is localized [25,51]. For the extrinsic use of dsRNAs as DC-targeting adjuvants, it is essential to appropriately design dsRNAs to guide them to the endosomes [25,26]. Several studies have demonstrated that the modification of dsRNAs facilitated them to gain access to endosomal TLR3 [25,26] (Table 1). ARNAX is 5′-GpC DNA-capped AU-rich dsRNA (120~140 bp) of a measles leader–trailer sequence, while PLGA-Riboxxim consists of PLGA and 100 bp GC-rich dsRNA (Table 1). In contrast, recent studies have suggested that ~424-bp dsRNA generated via in vitro transcription had TLR3-agonistic activity without transfection, which was similar to that of polyI:C [2,49]. The nucleotide segment was from the Chinese Sacbrood virus genome [54], implying that the nucleotide sequence would be a determinant in dsRNAs for internalization into DCs.

Hence, nucleic acids with appropriate formulation can enter endosomes with TLR3, making it possible to sense nucleic acids via TLR3 [2,25,26]. This non-inflammatory mode concerns the direct DC-priming mode [4,55]. In either mode, antigen-presenting DCs play a pivotal role in evoking the acquired immune system, where antigen-specific T cells proliferate.

## 5. TLR–TICAM-1 Pathway in DCs

TLRs are typical pattern-recognition receptors and recognize not only natural ligands but also chemically synthesized pattern molecules. Synthesized ones can be improved for safety and accessibility to the receptor. The combination of antigens and other molecules can also be considered for developing novel vaccines [11,56].

TLRs are expressed in multiple cell types, not just in DCs [51]. Therefore, it is impossible to exactly define the cell type from which TLRs transmit the signal output. TLRs other than TLR3 link to the MyD88 adaptor; yet, TLR3 and TLR4 recruit TICAM-1 [33,50]. As shown in Batf3^−/−^ mice, DC is essential for antigen-presenting capacity [17,18].

The TICAM-1 pathway is essentially involved in TLR3-specific signaling pathways [57,58], and CD141^+^ DCs cause TLR3-dependent cross-antigen presentation [18], suggesting that the TICAM-1 pathway is indispensable for cross-antigen presentation in this professional antigen-presenting DC subset [59]. Although the MyD88 pathway is also involved in TLR-mediated DC maturation, TLR3 does not link MyD88 [58], suggesting that the TICAM-1 pathway is crucial for the antigen presentation of cDCs by TLR3. CD141^+^ DCs tend to produce high amounts of IFN-λ in response to TLR3 ligands. Of note, mouse cDCs express TLR4 in addition to TLR3, and are capable of TLR4-driven cross-antigen presentation [18]. In human, CD1c+ DCs express TLR4, and TLR4-TICAM-1-mediated antigen presentation occurs in response to TLR4 ligands [18,29]. However, it remains unclear whether human TLR4-expressing cDCs cross-prime exogenous antigens as efficiently as CD141^+^ DCs. 

In other studies, the cytoplasmic RIG-I-like receptor (RLR), RIG-I/MDA5, has recognized dsRNA and/or triphosphate-capped viral RNA in cDCs and has activated cDCs for cross-antigen presentation [17,28,60]. This issue has been confirmed via the dsRNA adjuvant PLGA–Riboxxim [26] (Table 1). The route of s.c. administration may relieve the side effect [25,26]. polyI:C is transported from TLR3 endosomes to the cytoplasm to activate not only TLR3 but also MDA5 [26,61]. The activation of DCs by polyI:C is therefore complex, being associated with the TLR3–TICAM-1 and MDA5–MAVS pathways in whole-body cells, resulting in cytokinemia [62,63]. Furthermore, as other sentinels also exist in the cytoplasm, such as DEAD-box helicases, the possibility of their involvement in cDC cross-antigen presentation cannot be ruled out [63,64]. 

CD141^+^ DCs (a subset of cDCs that are XCR1 positive) preferentially express TLR3 and recognize dsRNAs [24]. One can predict by assessing blood cytokine/chemokine levels whether TLR3-targeted cDC activation is caused by inflammatory or non-inflammatory DC priming. The non-inflammatory adjuvant ARNAX can induce Th1 polarization and proliferation of antigen-specific CD8^+^ T cells despite no circulating inflammatory cytokines being detected in a mouse comparison study using polyI:C [61,65]. Similarly, in CD1c^+^ DCs (a subset expressing TLR4), MPLA has been found to induce Th1 polarization in a less-inflammatory manner, whereas LPS induces systemic inflammatory cytokines and IFN-I [34,35]. Immune activation and inflammation in vaccines can be separated via the selection and modification of adjuvants. 

In particular, cDCs directly sample microbe-associated molecular patterns (MAMPs) and induce antigen presentation [2,55]. The functional difference between viral dsRNA and polyI:C is notable. External dsRNAs derived from viruses rarely enter cells unless forcibly transfected, whereas polyI:C, consisting of heterogeneous double-stranded I:C, enters cells just by external addition and activates both TLR3 and systemic RLRs, resulting in RLR-mediated cytotoxicity. Thus, the polyI:C ligand can be classified as inflammatory, as being functionally similar to LPS for TLR4, and as CpG and an imidazoquinoline compound (R848 or Resiquimod) for TLR9/7. In contrast, the direct targeting of CD141^+^ cDC has been feasible without inflammation, as shown by the TLR3-specific adjuvant ARNAX since 2015 [25]. These non-inflammatory adjuvants are superior to other previous adjuvants in terms of safety (Table 1), but the suitability of adjuvants might vary on a case-to-case basis [26].

## 6. Modulation of Microenvironment by TLR3 Agonists

TLR3-specific agonists, in combination with IFN-I and COX2 inhibitors (p.o.), may provide some benefits in terms of therapeutic vaccine immunotherapy for use in the oncology field [66,67]. TLR3 agonists, when administered intravenously, promote the local infiltration of CTLs and inhibit the proliferation of regulatory T cells (Tregs), myeloid-derived suppressor cells (MDSCs), granulocytes, and other regulatory cells [66,67]. This therapy is referred to as a chemokine modulation regimen (CKM) [66]. It is usually used without concomitant tumor-associated antigens. It seems unlikely that this treatment without an antigen results in the proliferation of tumor-specific CD8^+^ T lymphocytes (CTLs). Nevertheless, intra-tumor CTL infiltration predicts favorable clinical outcomes in patients with multiple cancer types [68,69,70,71,72]. The clinical results are supported in mouse models, where TLR3 agonist + PD-1/L1 inhibitors enhance the antitumor response compared to the PD-1/L1 antibody only [73,74]. In contrast, the expansion of intra-tumor Tregs predicts poor prognosis [75,76]. 

Various factors that comprise the TME have been reported to be involved in tumor regression besides CTL infiltration [77]. cDC subsets are numerous, and some of which may be involved in the formation of the TME. cDCs tend to be tolerogenic in response to factors within the TME [78]. The TME can be altered based on the spatiotemporal phase of the cancer and the treatment [43,76]. This scenario has been tested in tumor-bearing mice and cancer patients, indicating that CKM has unique potential for use in reprogramming the local TME, as it selectively induces CTL attractants (but not Treg attractants) in tumor tissue and acts preferentially on the tumor (but not normal tissue) [77]. TLR3-specific agonists contribute to an improvement in the TME by reprograming the cell–cell network and promoting DC maturation. It is possible to safely apply these TLR3-specific agonists in combination with PD-1 inhibitors in an integrated manner [65,73]. 

In infection models, TLR3 agonists are considered to be highly effective as adjuvants in prophylactic vaccines to prevent life-threatening pneumonia of the host in influenza and severe acute respiratory syndrome (SARS) infections [10,11,56,79]. The safety barrier provided by the U.S. Food and Drug Administration is pretty high because the vaccine is usually intended for healthy individuals. Unlike cancer patients, healthy people do not have lesions corresponding to the TME. In situations where prophylactic vaccines are administered subcutaneously or intranasally, it is challenging to test vaccine efficacy by measuring the blood levels of cytokines/chemokines. T cell subsets may critically affect environmental factors. Mouse models with chronic infection or lesions such as in the TME might be used to verify another TLR3 effect on the microenvironment [80]. Otherwise, this TLR3 function on a local injected lesion would not have a meaningful impact in vaccination.

CTLs and Tregs are known to be attracted by different chemokines; CXCL9, CXCL10 (IP-10), and CXCL11 bind to CXCR3 expressed by CTLs [81], while CCL5 binds to CCR5 on effector cells such as CTLs and natural killer cells [82,83]. Meanwhile, CCL22 and CXCL12 attract Tregs and MDSCs via the receptors CCR4 and CXCR4 [61,83]. Treg and MDSC attractants may interfere with the cDC function and the chemokine network in vaccinations [84]. Again, it is notable that cDCs trap antigens at the site of infection or in the TME and migrate to their LNs, where cDCs are differentiated into a mature form to induce T cell cross-priming. Of note, macrophages do not migrate to the LNs.

## 7. Discussion

In vaccines, the definitions of antigen and adjuvant are a necessity. This section will discuss the differences in vaccine antigens required for infectious diseases in comparison with cancer, and the appropriateness of adjuvants depending on the antigen.

Infectious disease antigens can be predicted prior to infection, whereas cancer antigens cannot be predicted prior to disease onset. Therefore, no prophylactic cancer vaccine consisting of an antigen and an adjuvant has been established except cancer species expressing viral antigens (i.e., cervical cancer). Cancer arises by circumventing host immunity; it is extremely rare for cancer cells to evade the immune system and grow into a visible cancerous mass, which is seldom achieved via random genetic mutation, flotation, and selection [43]. This scenario is supported by the Darwinian view of cancer evolution [85].

For cancer cells to be recognized by immune cells, cancer antigen peptides must be properly presented on the cell surface. Abnormalities in molecules involved in antigen presentation, particularly in the expression of MHC molecules on cancer cells, are closely associated with therapeutic resistance [86]. IFNγ receptors and their signaling molecules are also intimately involved in the efficacy of immunotherapy [43,87], which was reinforced in immune checkpoint inhibitor trials [88,89].

These findings may be equally applicable to vaccines against intractable infectious diseases [80,90]. Microbial evolution frequently involves genomic variation and selection via immune surveillance. Antigen mutation and selection are driven by the contingent environment, including immune status. The efficacy of vaccines against infectious diseases can be predicted by neutralizing antibody titers and antigen-specific CTL induction levels.

Factors that influence the systemic immune status have been elucidated via studies into immunotherapies [91]. In addition to genetic pre-disposition and gut microbiota, lifestyle habits such as smoking, metabolism, caloric intake and obesity, infections, and regular medications, and environmental factors such as sunlight, air, and radiation are known to influence the dynamics of immunity [91]. 

Infectious disease vaccines should be superior in terms of immune specificity and memory. When designing a vaccine, it is important to carefully select the antigen and prioritize the induction of neutralizing antibodies that do not cause VDE and CTLs that are less likely to be exhausted. The quality and safety of antigens are crucial matters in addition to adjuvants for vaccines. The worsening of autoimmune diseases and difficulty in inducing memory immunity may be partly due to poor antigens. For vaccines with low efficacy, it is necessary to consider not only the adjuvant, but also the quality of the antigen. In the context of this approach, we mention the story of Nucleoside-modified mRNA vaccines (“mRNA vaccines”) against COVID-19. 

The mRNA vaccines against COVID-19, named BNT162b2 (Comirnaty, Pfizer/BioNTech) and mRNA-1273 (Spikevax, Moderna), passed FDA or European Medicines Agency (EMA) approval for the first time in 2020. They consist of two main components: 1. a nucleic-acid-modified mRNA encoding a spike (S) protein and 2. lipid nanoparticles (LNPs) that can efficiently deliver the mRNA into the cytoplasm [92]. These mRNA vaccines are very effective in inducing acquired immune responses, especially neutralizing antibodies that protect against SARS-CoV-2 infection [93,94,95,96] and promote humoral immunity. However, the approved corona mRNA vaccines cause fever in more than 50% of vaccinees and weak memory consolidation in lymphocytes, even after three doses, often exacerbating the underlying diseases. Specific CTL induction may further contribute to the prevention of COVID-19 severity [97,98,99,100], but little is known about which axis of innate immunity is involved in promoting the protective immune response.

Since empty LNPs can induce a variety of cytokines, including IL-1β, IL-6, GM-CSF, and many chemokines, most innate immune activation is likely dependent on the LNP component in the mRNA/LNP complex [101,102,103]. Clinical studies to date suggest that there is only a weak association between the severity of these adverse events and the magnitude of the antibody response, and little correlation between the T cell response and reactogenicity [104]. The MDA5-IFN-α signaling pathway has been implicated in BNT162b2 in mice to CD8+ T cell responses [105], but factors that activate the MDA5 pathway have not yet been identified [106]. Perhaps, LNPs or impurities are involved in the activation of this pathway. Based on this background, LNPs reduce the quality of mRNA vaccines in terms of poor adjuvanticity; it is hoped that vaccines will be established that not only reduce mRNA vaccine side effects (rarely, anaphylaxis and acute myocarditis or pericarditis) but also have safer, stronger, and more durable acquired immune responses. TLR3 adjuvants would be candidates for playing a role in improving the efficacy and safety of mRNA vaccines.

This suggests that a harmless adjuvant, if available, could be administered in combination with the antigen (either latent or tangible) to prevent infections. Vaccine compounds like LNPs rather than the modality of the antigen cause a low quality of immune modulation. Likewise, a vaccine to prevent cancer would be feasible with high quality once the antigen candidates are predictable. Any antigen will do to complete the vaccine for combination with the harmless adjuvant. 

For cancer vaccines, the questions are whether sufficient antigens can be delivered to dendritic cells and how the antigen will behave in the lesion microenvironment. The problem with cancer vaccines is that effective CTL induction is unlikely to occur even when large doses of the antigen are administered [107], which may be different from infectious antigens. In general, antigens in infectious disease vaccines are apparently of foreign origin, whereas cancer antigens are more closely related to self-antigens with a narrower range of mutations. If the effectors are mainly CTLs, which are required for reinvigoration, the vaccine formulation for appropriate antigen delivery to dendritic cells, rather than the tertiary structure of the antigen, is an important issue to consider. The appropriate specification of antigen and adjuvant will be critical in the development of vaccines for cancer and infectious diseases. 

## 8. Conclusions

The discrimination of self from non-self is an essential aspect in the conservation of species. The vertebrate provides the immune system for circumventing this self-conservation. The immune system consists of innate and acquired immunity, which is the recognition of non-self patterns by pattern-recognition receptors (PRRs) in the innate system leading to DC activation followed by acquired immune activation. The principle of pattern sensing is conserved across vertebrates, including humans. Innate pattern sensing links a sophisticated immune response in humans as an issue to be adapted to a vaccine. Hence, vaccines can be devised based on the principle of an immune response against refractory viral diseases.

Foreign nucleic acids are representative MAMPs. TLRs transduce the innate stimuli of MAMPs to immune effector output. Foreign nucleic acids are recognized in endosomes in DCs where nucleic-acid-sensing TLRs exist. In this sense, TLRs work as non-self PRRs. DCs consist of a variety of subsets, including pDCs, and a variety of cDCs. Each DC subset expresses unique repertoires of TLRs. Finally, specific subsets of cDCs induce cross-antigen presentation to proliferate antigen-specific CTLs. The production of antigen-specific neutral antibodies are in part dependent on the function of DCs. 

In the field of cancer vaccines, nucleic acid adjuvants help cDCs in inducing Th1 skewing and modulating the TME. CpG DNA and polyI:C were early candidates for nucleic acid adjuvants in therapeutic cancer vaccines. Currently, CpG ODN has been identified as a TLR9 agonist and polyI:C has been identified as a TLR3 and MDA5 agonist (Figure 1). Both induce strong inflammation (i.e., cytokinemia) when administered directly. Cytoplasmic sensors, the so-called sentinels, may further participate in the vaccine-mediated response. In this context, a TLR3-specific agonist (not spread over other sentinels) has been developed for a non-inflammatory adjuvant. In cancer vaccines, adjuvants should be designed to be less toxic and adapted to a therapeutic purpose. The TLR3-specific adjuvant has been successfully accomplished to establish non-inflammatory vaccines in mouse models. 

Knowledge of the antitumor vaccine adjuvants may apply to prophylactic vaccines for eradicating refractory infectious diseases. Vaccines should be superior in terms of immune specificity and the induction of memory. When designing vaccines, it is important to carefully select antigens and prioritize the induction of neutralizing antibodies that do not cause VDEs and CTLs that are less prone to exhaustion. Hence, the adjuvants should be selected in addition to antigens in vaccines against infection. The exacerbation of autoimmune diseases and the difficulty in inducing immune memory can be partly attributed to poor adjuvants. Vaccines with low efficacy need to be verified not only for the antigens but also for the adjuvants. In line with this endeavor, we will be able to develop harmless vaccines against cancer and infectious diseases. 

## Figures and Tables

**Figure 1 cells-12-01504-f001:**
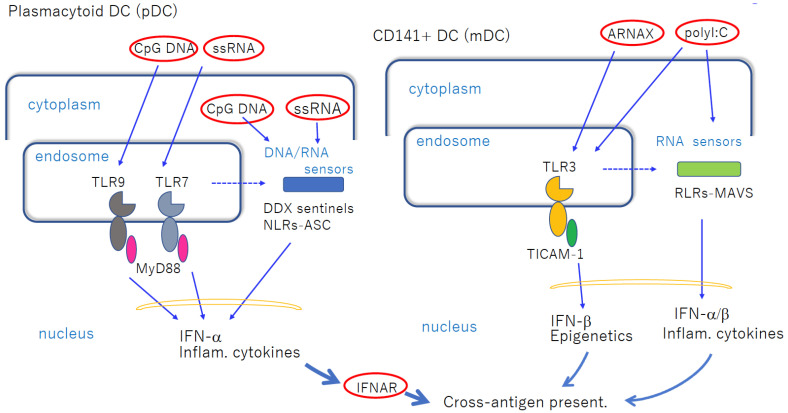
Nucleic-acid-sensing TLRs in pDCs and cDCs facilitate cross-antigen presentation. Left panel: pDCs express TLR7 and TLR9 in endosomes, which recognize imidazoquinoline and CpG DNA, respectively, when exogenously added. MyD88 transduces the signal to NF-κB activation to induce robust inflammatory cytokines and IFN-α. These mediators act on cDCs to induce cross-antigen presentation. If the ligands are transfected into cytoplasm, they are recognized using cytoplasmic sensors. pDCs do not present antigens per se. Right panel: CD141^+^ DCs express TLR3, and are increased via stimulation with dsRNA. Although blunt-ended viral dsRNA barely enters the endosome, ARNAX and PLGA-Riboxxim enter the endosome to activate TLR3 without transfection reagent. polyI:C activates both TLR3 and RLRs without transfection. Break lines indicate putative routes of nucleic acids to move from endosome to cytoplasm. The availability of this route depends on the cell types and properties of ligands. The properties of the transfection reagent affect the delivery of DNA/RNA. cDCs mature via direct TLR3-targeting without significant cytokine liberation in circulation.

**Table 1 cells-12-01504-t001:** Structure-defined dsRNA.

Items	ARNAX	PLGA-Riboxxim	NexaVant
dsRNA size(bp)	120~140	100	424
Structurally defined	yes	yes	yes
endosomal delivery	5′-DNA	PLGA adsorbed	Internal sequence
Target receptor	TLR3	TLR3/RIG-I	TLR3
Target cells	CD104+DC	CD104+, CD1c+DC	DC?
pathway	TLR3	TLR3/MAVS	TLR3/MAVS?
stability	high	high	high
GMP possibility	yes	yes	yes
Date of first publication	2015	2021	2023

## Data Availability

Available.

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
