# Peer review of "Two Modes of Th1 Polarization Induced by Dendritic-Cell-Priming Adjuvant in Vaccination"

_cells, 2023, doi:10.3390/cells12111504_

Round 1

Reviewer 1 Report

The manuscript, "Two modes of Th1 polarisation induced by dendritic cell adjuvant in vaccination", reviewed the nucleic acids as potential adjuvants and their roles and downstream effects in DCs. The article is well-organised and written concisely and informatically. Figure 1 summarises the essence of the article.

However, the authors left out two major comments the reviewer wants to point out. The key topic the reviewer wanted to see was the potential advantages and disadvantages of nucleic acids as vaccine adjuvants in mRNA or DNA vaccines. Since COVID-19, mRNA and DNA vaccines have been the current vaccine research and industry trend. Thus, assigning a section for the advantages and disadvantages of nucleic acids as adjuvants in mRNA/ DNA vaccine formulation is sensible.

The authors focus on the role, signalling pathways and their downstream effect of TLR3 and TLR7; however, there is no description for TLR8. TLR7 and TLR8 are similar in agonist and signalling pathways but different in their downstream effect. Therefore, it is crucial to describe the stimulation of TLR8 and the effect of its activation on DCs as a potential vaccine adjuvant.

Minor comments

1. The title needs to be clarified. The current title may mislead readers about dendritic cells as adjuvants of the vaccine.

2. The manuscript was well organised and easy to follow from section to section until it reached the “6. Discussion”. The reviewer had a hard time understanding the purpose of the section. The authors describe that the session would discuss the difference in vaccine antigens required for infectious diseases and cancer and the appropriateness of adjuvants depending on the antigen. However, the section does not get along with the other sections; no comparison of DCs’ roles between therapeutic and prophylactic vaccines and no comparison of the effect of TLR3- and/or TLR7-activated DCs on therapeutic and prophylactic vaccines. Thus, despite its very knowledgeable information, the reviewer put question marks for the “6. Discussion” section.

Author Response

Comment reply

Reviewer 1

  We appreciate the comments raised by the reviewers.

We have provided point-by-point reply to the comments.

Response to the reviewer comments’ part 1: “The key topic the reviewer wanted to see was the potential advantages and disadvantages of nucleic acids as vaccine adjuvants in mRNA or DNA vaccines. Since COVID-19, mRNA and DNA vaccines have been the current vaccine research and industry trend. Thus, assigning a section for the advantages and disadvantages of nucleic acids as adjuvants in mRNA/ DNA vaccine formulation is sensible.”

We would like to thank the reviewer for the productive comments. According to this and the final comments, We have revised the discussion section to reflect the findings on covid-19 mRNA vaccines. We discussed the advantages and disadvantages of nucleic acids as adjuvants in mRNA vaccine formulation. Thus, the discussion section was largely updated.

Response to the reviewer comments’ part 2: “there is no description for TLR8. TLR7 and TLR8 are similar in agonist and signalling pathways but different in their downstream effect.”

We would like to thank the reviewer for this inquiry. TLR8 express both pDCs and cDCs, and the expression levels are decreased during DC differentiation. The expression level is relatively high in monocytes. There are several reports describing negative regulation of TLR8 in the DC antigen-presenting function. This comment is also reflected in the Discussion section and page 5 line 185 (pDC and cDC section).

Minor comments

  1. Title was changed according to the comment.
  2. The reviewer gave us an important suggestion. We have extensively revised the Discussion section according to the comment.

Reviewer 2 Report

The authors aim to summarize the role of vaccine adjuvants in driving dendritic cell (DC) activation and polarization of immune responses towards cellular immunity. While the review is relatively easy to read, the focus and integration of the various sections is at times confusing. Specifically,

1. The authors state (Introduction) that their review will focus on vaccines for infectious diseases but the text contains multiple paragraphs/sentences related to cancer vaccines. It might make sense to select just infectious disease to keep the review focused.

2. The authors should also more clearly state (and rationalize) what type of adjuvants the review will focus on: virus-derived nucleic acids only or also other adjuvants.

3. The review does not mention pDC2, skin DCs/migratory DCs, other DC subsets that are potentially triggered dependent on which adjuvant and route of administration are used for vaccination.

Other comments:

4. In section 4, the authors state that TLR9 favors B-cell mediated antibody production. This seems contrary to reports showing CpG elicits T cell mediated immune responses.

5. Table 1 mentions CD104+ DC; the authors may want to add a few sentences in the text about CD104 to clarify the importance of this marker.

6. Also, it might be useful to expand table 1, if possible, to include information on adjuvants in clinically-approved vaccines.

7. There are two sections numbered “5”.

8. The authors state (Discussion) that no prophylactic cancer vaccines exist. However, cancer vaccines can be somewhat prophylactic when given in the adjuvant setting to prevent recurrence. Additionally, a vaccine to prevent cervical cancer (HPV vaccine) is given prophylactically.

9. The conclusion that foreign nucleic acids are recognized in endosomes through nucleic acid-sensing TLRs seems to contradict the statement on p.6 that “External dsRNAs derived from viruses rarely enter cells unless forcibly transfected”.

Minor editing required

Author Response

Reviewer 2

  We appreciate the comments raised by the reviewer. The text was corrected according to the comments except the comment #6.

  1. I agree to this comment.This issue was reflected in the initial paragraph of the introduction.

  1. This issue was also reflected in the revised introduction.

  1. We found this comment interesting. We focused cDC and pDC in this review. Otherwise, the point will be vague. Although this review did not intend to summarize a topic of dendritic cells, we mention the role of a variety of DC subsets in relevant sections. We found that DC specialists have joined the field of vaccinology and promoted the immunological aspect of vaccine function. Although DCs include a variety of subsets, we only focused cDC and pDC in association with TLRs agonist function.

  1. The reviewer is right. We reflected this issue in the section 4.

  1. CD141 represents an epitope of thrombomodulin, and named BDCA3. We mentioned this in the text.

  1. We regret that so far, no TICAM-1 adjuvant made of RNAs have been clinically approved.

  1. Corrected
  2. Except cancer with foreign antigens originated from infectious microbes, no prophylactic vaccine against cancer has been developed. We reflected this point in the section.
  3. Revewer is right. We corrected the sentence to “foreign nucleic acids are recognized in endosomes through nucleic acid-sensing TLRs when they reached successfully to endosomes”.